# Single-Cell Sequencing of Malignant Ascites Reveals Transcriptomic Remodeling of the Tumor Microenvironment during the Progression of Epithelial Ovarian Cancer

**DOI:** 10.3390/genes13122276

**Published:** 2022-12-02

**Authors:** Yiqun Li, Wenjie Wang, Danyun Wang, Liuchao Zhang, Xizhi Wang, Jia He, Lei Cao, Kang Li, Hongyu Xie

**Affiliations:** 1School of Life Sciences and Technology, Harbin Institute of Technology, Harbin 150000, China; 2Department of Biostatistics, School of Public Health, Harbin Medical University, Harbin 150000, China; 3Department of Gynecology, Women’s Hospital, School of Medicine, Zhejiang University, Hangzhou 360000, China; 4Clinical Research Center, Women’s Hospital, School of Medicine, Zhejiang University, Hangzhou 360000, China

**Keywords:** epithelial ovarian cancer, malignant ascites, single-cell sequencing, tumor microenvironment, targeted therapy

## Abstract

Epithelial ovarian cancer (EOC) is the main cause of mortality among gynecological malignancies worldwide. Although patients with EOC undergo aggregate treatment, the prognosis is often poor. Peritoneal malignant ascites is a distinguishable clinical feature in EOC patients and plays a pivotal role in tumor progression and recurrence. The mechanisms of the tumor microenvironment (TME) in ascites in the regulation of tumor progression need to be explored. We comprehensively analyzed the transcriptomes of 4680 single cells from five EOC patients (three diagnostic samples and two recurrent samples) derived from Gene Expression Omnibus (GEO) databases. Batch effects between different samples were removed using an unsupervised deep embedding single-cell cluster algorithm. Subcluster analysis identified the different phenotypes of cells. The transition of a malignant cell state was confirmed using pseudotime analysis. The landscape of TME in malignant ascites was profiled during EOC progression. The transformation of epithelial cancer cells into mesenchymal cells was observed to lead to the emergence of related anti-chemotherapy and immune escape phenotypes. We found the activation of multiple biological pathways with the transition of tumor-associated macrophages and fibroblasts, and we identified the infiltration of CD4^+^CD25^+^ T regulatory cells in recurrent samples. The cell adhesion molecules mediated by integrin might be associated with the formation of the tumorsphere. Our study provides novel insights into the remodeling of the TME heterogeneity in malignant ascites during EOC progression, which provides evidence for identifying novel therapeutic targets and promotes the development of ovarian cancer treatment.

## 1. Introduction

Epithelial ovarian cancer (EOC) is the most deadly gynecological malignancy [1]. Approximately 300,000 new cases of EOC are estimated to occur annually, and nearly all women experience recurrence, with a 5-year survival rate of approximately 48% [2,3]. It is important to understand the potential mechanisms of EOC from diagnosis to relapse so as to achieve early intervention to extend survival.

EOC displays a high degree of intratumor heterogeneity, which is manifested in cancer cell clonal heterogeneity and stromal microenvironmental diversity [4]. Cancer cells and tumor microenvironment (TME) cells with plasticity change from one cell state to another to enable essential tumor characteristics, such as recurrence [5]. Symptomatic malignant ascites is a typical symptom of EOC. More than 30% of patients with EOC had peritoneal ascites at diagnosis and nearly all developed ascites at recurrence [6]. Malignant ascites provides a distinctive environment comprising various active cells and biological components, which facilitates cancer cell adhesion, migration, and invasion. Bulk ascites transcriptome analysis indicated that the process of epithelial–mesenchymal transition (EMT) affects the molecular and biological characteristics of tumor cells and leads to ovarian cancer (OC) progression [7]. However, bulk-cell sequencing approaches have not considered the possibility of multiple cell states co-existing in one tumor, which are often confounded by non-cancer cells [8]. Single-cell RNA sequencing (scRNA-seq) is an innovative approach for analyzing the regulation, differentiation, and connection of individual cells and provides a powerful means to characterize tumor intercellular heterogeneity [9]. Max et al. conducted scRNA-seq on 3114 cells collected from four OC ascites samples to describe the cell composition of OC ascites [10]. To further evaluate the heterogeneity of malignant cells among patients with high-grade serous ovarian cancer (HGSOC), Izar et al. performed a landscape scRNA-seq analysis on 22 ascites samples from 11 patients and found that immunoreactive and mesenchymal subtypes of HGSOC were actually caused by the infiltration of abundant immune cells and fibroblasts instead of tumor cells [11]. This study enhanced our cognition of HGSOC and provided critical information to elucidate the role of the TME in ascites. Previous studies on scRNA-seq in OC ascites mainly focused on the landscape of cell composition and its interaction with the TME. However, the underlying specific cellular and molecular mechanisms of malignant ascites in promoting the progression of OC have not yet been elucidated. 

In this study, we postulated that EOC progression was intimately linked to accumulated cell phenotype transition and genetic expression alteration mediated by ascites. We explored the dynamic changes in the specific characteristics of cell subpopulation profiles and the variation of corresponding biological functions from diagnosis to recurrence based on scRNA-seq, which might contribute to the knowledge of disease development and provide early intervention to extend the survival of patients with EOC.

## 2. Materials and Methods

### 2.1. Study Subject

The present single-cell analysis dataset was obtained from the Gene Expression Omnibus (GSE 146026). Raw data are available via the Broad Institute Data Use Oversight System (https://duos.broadinstitute.org/#/home, accessed on 1 March 2021). GSE 146026 included 22 samples from 11 patients detected by Smartseq2 and/or by 10x platforms, and samples that were sorted according to EPCAM^+^CD24^+^ upon detection (samples 5.2, 7, 8, 9, 10 and 11) and received initial treatment (samples 4 and 5.2) were excluded [11]. A total of 4680 cells were extracted from a set of five patients (three diagnoses and two relapses), namely, samples 1, 2, 3, 5.1 (represented by sample 5), and 6. Samples 3, 5, and 6 were collected at the time of diagnosis. Samples 1 and 2 were collected at recurrence. All ascites samples were detected using the 10X Genomics platform. The characteristics of these patients are listed in Table 1. Bulk RNA-seq data and the corresponding clinical information of patients with OC were obtained from The Cancer Genome Atlas (TCGA, http://www.cancer.gov/tcga, accessed on 1 March 2021) and Gene Expression Omnibus (GEO, http://www.ncbi.nlm.nih.gov/geo/, accessed on 1 March 2021) databases. Bulk RNA-seq data retrieved from multiple databases were integrated using the Combat method with the “sva” package from the Python platform. All public data used in this study are described in Appendix A.

### 2.2. ScRNA-seq Data Processing

Further scRNA-seq data analysis was performed using Scanpy 1.5 in Python 3.6 (‘tf115’:conda). Genes were first filtered on the basis of their expression in at least three cells, with each cell requiring at least 200 genes to be expressed. Mitochondrial read counts were collected using the “var_names.str.startswith(‘MT-’)” function. The gene expression was normalized in each cell by log-transformation according to the total read count. Afterward, central scaling was performed across all batch data using default parameters. The top 2000 highly variable genes (HVGs) were selected for downstream analysis using the “sc.pp.highly_variable_genes” function.

### 2.3. ScRNA-seq Data Analysis

The HVGs were subjected to principal component analysis (PCA). The principal components (PCs) representing the most variation in the dataset were chosen using an elbow plot. To recognize cell clusters, these PCs were used as edge weights in an unsupervised graph-based clustering. The PCA reduction dimension was used to calculate the similarity among cells. A closer sample distance indicated a more similar cellular gene expression pattern. T-distributed stochastic neighbor embedding (*t*-SNE) was used for the visualization of the cell clusters and the evaluation of the performance of batch correcting. Cluster annotation was determined on the basis of the expression levels of classical markers. We utilized the Wilcoxon rank sum test to identify the differentially expressed genes (DEGs) of each cell cluster using the “sc.tl.rank_genes_groups” function with default parameters. A *p* value < 0.05 (adjusted using the Bonferroni method) and log2 fold change > 1 were regarded as the cutoff values. For sub-clustering analysis, we applied the same procedure of dimensionality reduction, clustering, and identifying DEGs.

### 2.4. Batch Correction and Evaluation

The sample collection time and single-cell sequencing time were completely different among these five patients [11,12], and a strong batch effect was potentially present across different samples. Here, we utilized an scRNA-seq data batch-correcting algorithm named “DESC” [13], an unsupervised deep learning algorithm, to integrate different batches of scRNA-seq data while maintaining cell type separation. Using the iterative optimization of the clustering objective function, the DESC algorithm can balance biological and technological variances across clusters and eventually diminish the variation of batch effect. Then, we used two evaluation metrics, namely, “KL divergence” [13] and “silhouette score” [14], to verify the performance of DESC methods for batch variation elimination and the randomness with which cells from different batches are mingled together in each cluster. A smaller KL divergence or higher silhouette score implies improvement in batch mixing and a more effective batch variance elimination. We calculated the KL divergence and silhouette score for raw data (before batch removal) and data corrected by DESC algorithms, and we created box plots of the final silhouette score and KL divergence.

### 2.5. Pseudotime Analysis

To explore cell-state transitions, scRNA-seq trajectories were performed using the Monocle2 package. DEGs over the pseudotime among cluster cell transitions were generated using the “differentialGeneTest” function (q-value < 0.1). The “DDRTree” was conducted for dimension reduction and the visualization of trajectories. The minimum spanning tree was plotted on cells using the “plot_cell_trajectory” function.

### 2.6. Pathway Analysis and Functional Annotation

KEGG pathway enrichment analysis and GO functional annotation were performed using the Kyoto Encyclopedia of Genes and Genomes (KEGG) (http://www.genome.jp/kegg/, accessed on 15 April 2021) and Gene Ontology (GO) (http://geneontology.org, accessed on 15 April 2021) databases, respectively, to reveal the biological functions of DEGs. Pathways and GO terms with calculated q-values ≤ 0.05 were considered significant.

## 3. Results

### 3.1. Batch Correction

The DESC algorithm was used to integrate five heterogeneous ovarian cancer ascites scRNA-seq samples and refine clusters iteratively. This self-learning procedure can generate satisfactory feature representation and promote cell clustering by progressively eliminating the effect of batch variation. According to the visualization of the t-SNE, due to the batch effect between the different samples and the experimental platforms, many cell clusters were formed using the raw data before removing the batch effects. However, these cell clusters were not consistent with actual cell types (Figure 1a,b). Moreover, the same cell type from different batches could not be clustered under the influence of the batch variation, which confused the biological variation of interest during data integration. DESC was an effective method for removing batch effects and resulted in a relatively clearer cell clustering after batch correction (Figure 1c,d). As shown in Figure 1e,f, DESC significantly improved the silhouette scores (the median was 0.32) compared with the raw data (median = 0.00; *t*-test *p* < 0.05) and strikingly decreased the KL divergence (median = 0.86 vs. 0.71; *t*-test *p* < 0.05), thereby demonstrating the ability of DESC to accurately process the data-compiled unwanted batch effect for accurate downstream analysis.

### 3.2. A Single-Cell Atlas in Ascites of EOC

A total of 4680 cells of ascites were detected using 10X Genomics from five ovarian cancer patients; 2662 cells were obtained from patients 3, 5, and 6 at diagnosis, and 2018 cells were collected from patients 1 and 2 at recurrence. As shown in Figure 1d, six distinct major clusters (clusters 0–5) were characterized. According to the set of classical cell markers in Table 2, cluster 1 was classified as cancer cells (marked by PAX8), with 278 and 528 cells in diagnostic and recurrent samples, respectively. Cluster 0 was classified as macrophages (2104 cells, 44.96%, marked by CD68). Cluster 2 was classified as fibroblasts (763 cells, 16.30%, marked by COL1A2). Cluster 3 was classified as T cells (446 cells, 9.53%, marked by CD2). Cluster 4 was classified as B cells (425 cells, 9.08%, marked by CD79A). Cluster 5 was classified as dendritic cells (136 cells, 2.91%, marked by CD83) (Figure 2a–c). In the diagnostic samples, macrophages were the most abundant. In the recurrent samples, cancer cells, macrophages, and fibroblasts accounted for 83.89% of the cell population. As expected, compared with the diagnosis samples, the recurrent samples exhibited a relatively greater percentage of cancer cells and fibroblasts. Table 2 shows the number and proportion of these cell types in the samples obtained at diagnosis and recurrence.

### 3.3. Intrinsic Tumor Cell Subpopulations of Ascites

To discuss the heterogeneity of malignant cells of the EOC ascites in both diagnostic and recurrent samples, a total of 806 cancer cells were grouped into four subpopulations following single-cell sub-clustering analysis (Figure 3a,b). We found that subgroups S0–S3 expressed EPCAM, a well-known epithelial malignant marker, to varying degrees. Compared with the subgroups S0, S2, and S3, subgroup S1 had the highest EPCAM expression (Figure 3c,d). These results suggest that cancer cells in subgroups S0, S2, and S3 may not have sufficient properties of epithelial cells. To explore the biological status of these malignant subpopulations, we performed functional enrichment analysis based on differentially expressed genes. The results illustrated that multiple classical oncogenic pathways, including cell adhesion molecular, angiogenesis, myogenesis, transforming growth factor beta (TGF-β) signaling pathway, chemokine signaling pathway, IL-2 STAT5 signaling pathway, interferon (IFN) alpha response, and interferon gamma response, were activated from S1 to S0, S2, and S3 during EOC progression. All of these pathways were associated with cancer development and progression (Figure 3d).

We hypothesized that the process of EMT in ovarian cancer cells mediated phenotypic changes and related intra-cluster heterogeneity [15]. To determine the differential functional outcomes of cell subpopulations, we evaluated the expression of multiple epithelial–mesenchymal transition signatures [16] across clusters and investigated the potential biological characteristics as a function of markers for epithelial and stromal cancer cells in the literature. Some epithelial markers, such as EPCAM, CLDN4, KRT18, CDH1, OCLN, and CD24, were highly expressed in S1. However, S0, S2, and S3 exhibited markedly higher expressions of mesenchymal markers (CDH2, S100A4, MMP2, WNT5A, and ZEB1) (Figure 3e). Cancer cells highly expressing EPCAM were harbored in ascites samples obtained at diagnosis. Cancer cells highly expressing CDH2 were identified in the ascites samples obtained at recurrence. Heatmap analysis also showed that genes changed dynamically from S1 to S0, S2, and S3 (Figure 3f). Then, we performed the pseudotime analysis along the trajectory of S0–S3 according to gene expression profiles and demonstrated a differentiated process from S1 to S0, S2, and S3 (Figure 3g). S0, S2, and S3 may have originated from S1 with the progression of EOC. All results supported our hypothesis that EMT mediated the phenotype transition of malignant cells in EOC and certified that S1 represented epithelial ovarian cancer cells, whereas S0, S2, and S3 represented mesenchymal ovarian cancer cells that were not sufficiently identified in the previous ascites single-cell study of EOC.

To identify novel markers of these mesenchymal EOC cells, we further analyzed the DEGs and the biological functions of related pathways in S0, S2, and S3. These cells highly expressed the integrin family genes associated with the cell adhesion molecule pathway in contrast to epithelial tumor cells, such as ITGA2, ITGA3, ITGA5, ITGA6, and ITGAV, which were among the top-ranking DEGs. To further verify whether these integrin genes, highly expressed in mesenchymal tumor cells, were associated with the prognosis of OC, Kaplan–Meier survival curves and a log-rank test were used to assess the performance of each gene signature in the TCGA and GEO meta datasets. Patients were divided into high- and low-expression groups according to the median expression of each gene. The survival times of the high- and low-expression groups of integrin genes, except ITGAV, were statistically significant (Figure 3h). Among these selected genes, ITGA5 expression showed the most significant difference in the overall survival; a lower expression of the ITGA5 gene was correlated with a greater overall survival and better prognosis (log-rank *p* = 0.0025; HR = 1.83 (1.23–2.71)). Together, these results suggested that the transition of mesenchymal subtype cells with a high expression of integrin genes typically signified an invasive nature and worse clinical prognosis.

### 3.4. Distinct Subgroup in Mesenchymal Cancer Cells

Although the S0, S2, and S3 subgroups had stronger mesenchymal properties than epithelial cancer cells, S2 and S3 also specifically expressed genes in other related biological pathways compared with S0. We further analyzed the DEGs among the mesenchymal cells highly expressing ITGA5 in S0, S2, and S3. Some specific markers of S2 were found to be related to immunity, such as MUC16 and TNFRSF11B (Appendix A). Functional annotation also indicated that S2 might be related to host immunity (Figure 3e). Subgroup S3 specifically expressed ABCG2, MDR1, MRP1, and GST (Appendix A), which are associated with the chemotherapy resistance-related pathway (Figure 3e). The above observations further indicated the heterogeneity of mesenchymal tumor cells.

### 3.5. Identification of M2 Tumor-Associated Macrophages

To further characterize the macrophage clusters, we re-clustered the 2104 macrophages into two subgroups on the basis of t-SNE analysis (Figure 4a). Then, we inspected the expression of canonical marker genes of two macrophage subtypes, such as CD16 for the M1 subtype and CD163 for the M2 subtype (Figure 4c). We observed the changes in the proportions of M1 and M2 macrophages from the diagnostic samples to the recurrent samples. Unexpectedly, M1 was mainly concentrated in the diagnostic samples, and M2 was mainly distributed in the recurrent samples (Figure 4b). Pseudotime analysis further demonstrated a differentiated trajectory, i.e., M2 originated from M1 macrophages with the progression of EOC (Figure 4d), which is consistent with a previous study indicating the pro-cancer role of M2 [17]. The important pathways involved in macrophage function activation, such as the tumor necrosis factor (TNF) signaling pathway, NOD-like receptor signaling pathway, and nuclear factor kappa-B (NF-κB) signaling pathway, were regulated differently between the M1 and M2 macrophage subtypes in our pathway enrichment analysis (Figure 4e). 

### 3.6. Activated Cancer-Associated Fibroblasts Express Tumor Marker Genes

We then focused on the non-immune components, such as fibroblasts in the TME. Many studies have demonstrated that cancer-associated fibroblasts (CAFs) play an important role in the progression of cancer [18]. We identified 763 fibroblast cells. Two separate clusters were defined according to the similarities between each cluster and the expression of key markers in our single-cell data subcluster analysis (Figure 5a,b). Subgroup S0 was clustered in the early diagnostic samples, whereas a higher proposition of the S1 subgroup was found in the recurrent samples. The general stromal genes VIM and COL11A1 exhibited a strong and universal signal, although certain markers exhibited distinctive phenotypic diversity across these two subgroups (Figure 5c). ACTA2, which encodes the protein product α–smooth muscle actin, was abundant in the S1 subgroup [19]. Thus, we used ACTA2-positive CAFs to represent the S1 subgroup and used ACTA2-negative CAFs to represent the S0 subgroup. These ACTA2-positive CAFs had a significant positive correlation with the EMT-related transcription factors (such as SNAI1, TWIST1, and ZEB1) in several TCGA studies [20,21], suggesting that EMT progression might be a potential source of ACTA2-positive CAFs. Then, we conducted gene enrichment analysis on the top DEGs in each CAF subtype. Not surprisingly, the extracellular matrix, cell adhesion, and angiogenesis-related pathway were enriched in the ACTA2-positive CAF subtypes (Figure 5d). 

Both advanced EOC cells and activated CAFs were highly enriched in the cell adhesion molecule-related pathway, which might suggest that tumor cells and CAFs often aggregate into a multicellular spheroid formation in ascites. On the other hand, some genes that regulate extracellular matrix (ECM) formation and components, such as CREB3L1 and PLAGL1, had relatively high abundance in ACTA2-positive CAFs (Figure 5e), indicating that the activation of ECM is crucial for CAF differentiation. The results of pseudotime analysis for fibroblasts were unsurprisingly concordant with the activation states of CAFs, starting at the S0 subgroup and progressing toward S1 along the axis (Figure 5f).

### 3.7. CD4^+^CD25^+^ T Regulatory Cells Showed Infiltration Features

The involvement of immune cells in the TME in tumor progression is complex. We investigated the heterogeneity of infiltrating T cells in these EOC ascites samples from diagnosis to recurrence based on single-cell re-cluster analysis. A total of 446 T cells were subtyped into four subgroups (Figure 6a,b). According to specific marker gene expression, we identified CD8^+^ T effector cells (marked by GZMA), CD8^+^ T naïve cells (marked by CD8A), CD4^+^ T conventional cells (marked by CCR7), and CD4^+^ CD25^+^ T regulatory cells (marked by FOXP3) (Figure 6c). Pathway enrichment analysis of differentially expressed genes further confirmed the differentiation of T lymphocyte subtypes, which indicated an unbiased cell subgroup representation (Figure 6d). CD8^+^ T effector cells were expressed in the regulation of the MHC II biosynthetic process and regulation of interleukin-17 production. CD8^+^ T naïve cells highly expressed genes related to the regulation of inclusion body assemblies. CD4^+^ T conventional cells were highly enriched in genes related to the regulation of T-cell receptors and NF-κB signaling. Encouragingly, we identified CD4^+^ CD25^+^ T regulatory cells for the first time in EOC ascites samples in a single-cell study. These cells highly expressed other related markers, such as CTLA4, GITR, and NRP1 (Figure 6e). The top enriched terms or related pathways for CD4^+^ CD25^+^ T regulatory cells included regulation factor of T-cell differentiation, amino-acid substrates, and metabolism of glucose. RAGE receptor binding and Wnt protein, which were previously identified as participants in FOXP3^+^ Treg regulation, were also included.

Surprisingly, we noticed that the integrin cell adhesion molecule was also enriched in the top terms or related pathways for CD4^+^ CD25^+^ T regulatory cells. This result is consistent with our findings in the abovementioned tumor cell subcluster analysis that EOC ascites samples at the stage of progression contained more mesenchymal tumor cells highly enriched in integrin. Such consistency indicates that integrin expression on CD4^+^ CD25^+^ T regulatory cells promotes immune infiltration and impels Tregs to aggregate around tumor cells, thereby causing a local immunosuppressive state via some possible mechanisms.

### 3.8. Dendritic and Treg Cells Both Expressed IDO-Related Genes

Lastly, we focused on dendritic cells in ascites samples with EOC. The dendritic cells, considered to be the most important antigen-presenting cells (APCs) in vivo, are the only APCs that activate the initial type T lymphocyte [22]. We found that CD80, CD86, and IDO1 were highly expressed in dendritic cells (Figure 6f). CD80/CD86 can combine with CTLA4, which was highly expressed on the surface of CD4^+^ CD25^+^ T regulatory cells. This interaction between dendritic cells and T regulatory cells can activate indoleamine 2,3-dioxygenase (IDO) in dendritic cells, thereby reducing free tryptophan and decreasing the activation of T effector cells to promote immune escape. We found that GCN2 and AHR were also up-regulated in CD4^+^ CD25^+^ T regulatory cells (Figure 6g).

## 4. Discussion

Nearly all patients with EOC will experience recurrence [3], and inherently high intratumor heterogeneity in the TME is an important factor for ineffective treatment [23]. Bulk sequencing averages the transcriptome signals of cell mixtures and, thus, hinders the evaluation of cell heterogeneities. In this study, we conducted a comprehensive scRNA-seq expression analysis of five EOC ascites samples. We analyzed the heterogeneity of the TME in EOC ascites and described the key molecular differences between the stages of diagnosis and recurrence. We identified novel remodeling of cell subtypes and related pathways in the TME, along with tumor progression.

In the current study, we performed batch correction using “DESC”, a deep neural network and unsupervised embedding cluster analysis method, which gradually reduced the effect of batch variation by iteratively optimizing a clustering objective function under the condition that true biological differences were greater than technical variations across batches. The evaluation metrics, KL divergence and silhouette coefficient, showed that the compound batch effect was effectively corrected by the DESC algorithm for these multi-batch, integrated scRNA-seq data. After batch correction, we identified six cell types of EOC, namely, cancer cells, macrophages, fibroblasts, T cells, B cells, and dendritic cells. These results are consistent with those obtained in Izar’s study [11], thereby demonstrating the accuracy and stability of our method. Then, we performed a more thorough analysis and revealed that the TME underwent substantial changes in terms of cell phenotype and biological characteristics during the process of tumor progression. In the recurrent ascites, we found a large number of tumor cells with mesenchymal properties, and we identified a dominant proportion of M2 macrophages and activated tumor-associated fibroblasts. In addition, we discovered the infiltration of CD4^+^CD25^+^ T regulatory cells in recurrent ascites samples. These changes in the single-cell landscape in the malignant ascites TME allowed the exploration of the detailed mechanisms underlying tumor progression. 

These malignant tumor cell subtypes in recurrent samples displayed different phenotypic characteristics and biological functions compared with ascites tumor cells obtained at the diagnosis stage. One significant feature of these phenotype cancer cells was their high expression of mesenchymal markers, such as CDH2, S100A4, MMP2, WNT5A, and ZEB1. Pseudotime analysis further confirmed our hypothesis that these tumor cells with mesenchymal properties were derived from EOC cells through the EMT process. The differentially expressed genes of these mesenchymal tumor cells were enriched in the pathways associated with cancer carcinogenesis and progression, including epithelial–mesenchymal transition, interferon gamma response, TGF beta signaling pathway, cell adhesion molecular, mTORC1 signaling pathway, and PI3K/Akt signaling pathway, in line with the results of previous in vitro studies [24,25]. Transforming growth factor-β (TGF-β) is an inducer of the EMT process [24]. Several studies have documented that fibroblasts became CAFs in the TME after activation by TGF-β, and that they are involved in the formation of the cancer vasculature for tumor invasion and metastasis [26]. The PI3K/Akt/mTOR pathway is typically activated in cancer tissue and can maintain tumor growth; inhibition of the PI3K/Akt signaling pathway has been proposed as a therapeutic target [25]. IFN-γ is a crucial effector cytokine for tumor immunological rejection. Some researchers have shown that IFN-γ/IFN-γR signaling blockage can damage the immune system’s rejection of malignancies; the loss of IFN-γ produced by T cells can promote tumorigenesis and persistence [27].

One intriguing finding was that mesenchymal tumor cells also began to differentiate into distinct subtypes with different functions after acquiring and stably maintaining mesenchymal malignancy. The S2 subtype of mesenchymal tumor cells overexpressed MUC16 and TNFRSF11B, which might lead to immune resistance. MUC16 was expressed on the surface of OC cells and suppressed the interaction between OC and NK cells, causing them to avoid host immunity [28]. TNFRSF11B encodes a cytokine named osteoprogerin (OPG), which competitively binds with TRAIL to induce apoptosis of tumor cells [29]. In addition, some of the tumor cells overexpressed the ABCG2 that belongs to the ATP-binding cassette (ABC) transporter superfamily, which can pump chemotherapy drugs, such as paclitaxel, cyclophosphamide, and gemcitabine, out of tumor cells, thereby making the cells resistant to chemotherapy [30]. GST can increase the activity of intracellular detoxification enzymes and inhibit the response of tumor cells to chemotherapeutic drugs [31]. Our results indicated that the transformation of epithelial cancer cells into mesenchymal cells might lead to the emergence of related anti-chemotherapy and immune escape phenotypes. 

The higher proportion of M2 macrophages and ACTA2-positive CAFs in recurrent ascites samples further indicated the malignant transitional direction of the TME. Macrophages, CAFs, and cancer cells all worked and interacted with one another in concert to modulate the ECM within the TME. The activation of M2 macrophages and CAFs can be triggered by cytokines secreted from cancer cells, e.g., TGF-β, and multiple receptor tyrosine kinase signaling ligands, such as FGF and PDGF, while they can also be stimulated to secrete EGF, thereby enhancing the invasion of EOC [32]. Some in vitro studies indicated that CAFs preferentially induced the activation of NF-κB signaling for the transformation of M0-macrophages into a tumor-promoting (M2) phenotype [33]. CAFs and tumor-associated macrophages (TAMs) also produce a metalloproteinase named MMP2 to remodel ECM production and composition, thereby resulting in tumor cell invasion, progression, and metastasis, while supporting angiogenesis [34].

Immune cells undoubtedly play a significant role in the TME. The processes behind OC’s immune-suppressive environment are of particular concern. We identified and characterized the CD4^+^CD25^+^ Treg cells for the first time in a single-cell study based on ascites samples of EOC. As expected, the presence of these cells usually portended tumor progression and poor prognosis because of their immunosuppressive effect [35]. The proliferation of CD4^+^CD25^+^ Treg cells can be stimulated by inhibitory cytokines, such as IL-10 and TGF-β secreted from tumor cells within the TME, and they directly inhibit the activation of CD8+ effector T cells and restrict their IL-2 production. CD4^+^CD25^+^ Treg cells can also affect the function of dendritic cells by downregulating the activity of NF-κB, thereby significantly reducing the DC-mediated tumor cell-killing effect [36]. Chang’s study revealed that CD4^+^CD25^+^ Treg cells in the TME might rely on granzyme B and perforin to eliminate CD8+ effector T cells, resulting in a loss of function to help tumor cells escape from the toxic effect of CD8+ effector T cells [37]. Furthermore, the up-regulated expressions of IDO and tryptophan metabolism pathway-related genes, such as IDO1, GCN2, and AHR, in dendritic cells and Treg cells have been reported. IDO is an essential endogenous immune suppression factor that converts the tryptophan of T effector cells to generate kynurenic acid [38]. Exhaustion of tryptophan can activate GCN2 to encode the eukaryotic translation initiation factor 2 alpha (eIF2α) kinase 4, which can cause eIF2α phosphorylation and lead to the inhibition of protein translation [39]. Kynurenic acid is an endogenic ligand of ary1 hydrocarbon receptor (AHR) [40]. The exhaustion of tryptophan and accumulation of kynurenic induce the cell-cycle arrest of T effector cells, ultimately increasing their apoptosis and causing immunosuppressive effects.

Furthermore, our study revealed that the cell adhesion molecular pathway mediated by the integrin family was enriched in mesenchymal malignant cells, CAFs, and Treg cells. Among these integrin genes, the expression of ITGA5 was the most impressive. TCGA and GEO meta-data related to OC also indicated that a high expression of most integrin genes might result in a poor prognosis in terms of overall survival. Integrin is a crucial modulator that allows cancer, immune, and stromal cells to adhere to the ECM for tumorsphere formation [41]. Such a finding reveals a possible mechanism underlying the following functions of the multicellular tumorsphere: enhancing the communication among tumor, stromal, and immune cells; amplifying biological signals; and promoting tumor cells to acquire and maintain malignant phenotypes and local immunosuppressive status.

The main limitation of this study was its small sample size. The number of cells detected by scRNA-seq in ascites was lower than that in tissue samples, which hindered further subdivision of the specific cell types to comprehensively identify relevant pathways and potential therapeutic targets. We explored the potential mechanism of the progression of EOC using ascites samples, whereas we only validated their relevance to prognosis using the bulk data; thus, large prospective studies are needed for further validation.

## 5. Conclusions

Our study presented a single-cell transcriptomic remodeling of EOC malignant ascites, where multiple tumor progression-associated biological processes were found to be activated. These changes in the single-cell landscape highlight the heterogeneity and complexity of the TME and provide new perspectives for targeting a wider TME cell population, such as CAFs and TAMs. However, their roles in regulating the tumor immunologic microenvironment associated with the development of cancer need to be further studied.

## Figures and Tables

**Figure 1 genes-13-02276-f001:**
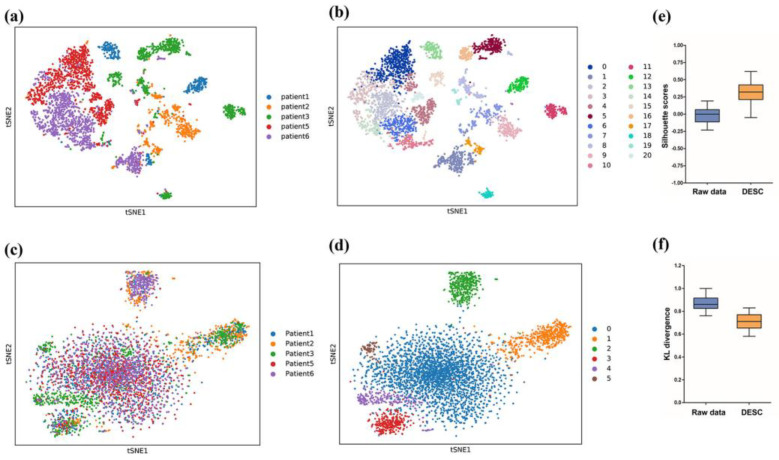
Batch correction and evaluation of multi-batch, integrated scRNA-seq data collected from ascites samples of patients with EOC. (**a**,**b**) T-SNE view of ascites samples before DESC correction, colored by batch ID (**a**) and cell clusters (**b**). (**c**,**d**) T-SNE view of ascites samples after DESC algorithm correction, colored by batch ID (**c**) and cell clusters (**d**). (**e**,**f**) Box plot of silhouette scores (**e**) and KL divergence (**f**) in raw data and data after DESC correction.

**Figure 2 genes-13-02276-f002:**
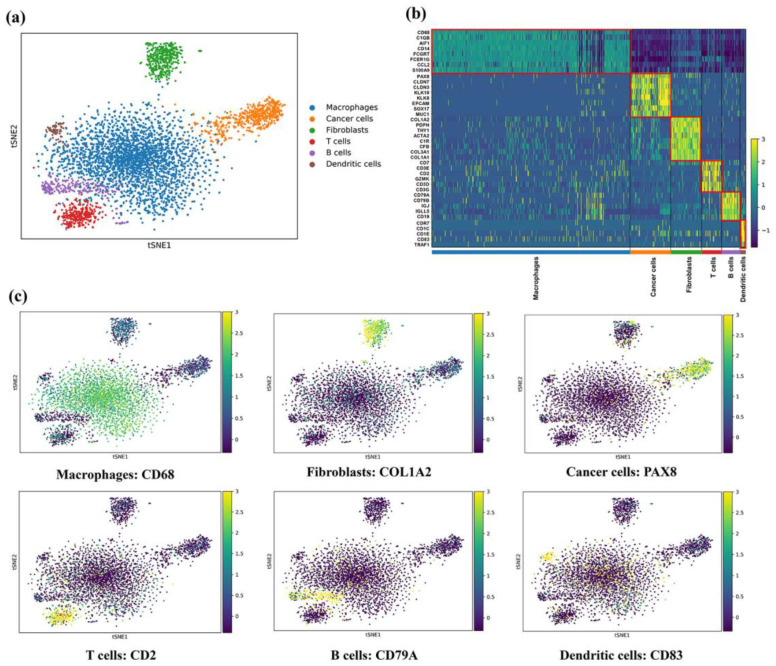
Overview of single-cell landscape for malignant ascites of EOC. (**a**) 4680 single-cells and six cell types were identified, colored by cell type. (**b**) Heatmap of marker genes for each cell type. (**c**) T-SNE map showing canonical markers for six cell types: CD68 for macrophages; COL1A2 for fibroblasts; PAX8 for cancer cells; CD2 for T cells; CD79A for B cells; CD83 for dendritic cells.

**Figure 3 genes-13-02276-f003:**
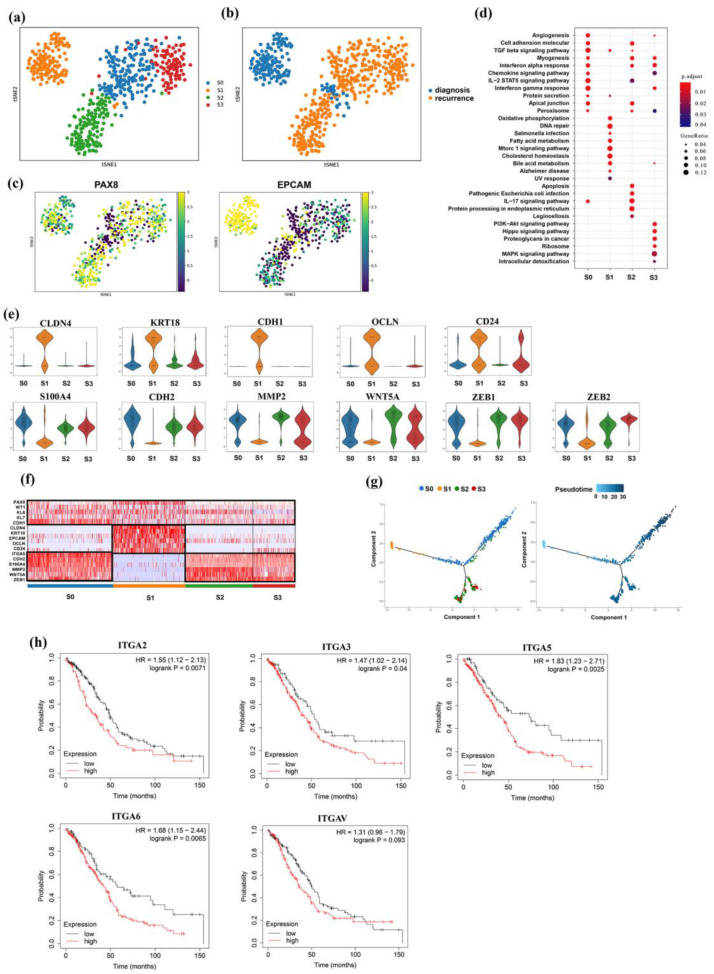
Identification of mesenchymal cancer cells in the progression of EOC. (**a**,**b**) T-SNE view of four subgroups of cancer cells, colored by subgroups (**a**) and sample groups (**b**). (**c**) The expression of PAX8 and EPCAM in the t-SNE map of diverse cancer cell subgroups. (**d**) Differential activated pathways among four subgroups of cancer cells. (**e**) Expression levels of epithelial mesenchymal transition markers in each cancer cell subgroup were plotted in violin plots. (**f**) Heatmap of representative markers for epithelial and mesenchymal cancer cells. (**g**) Pseudotime analysis demonstrates a differentiated trajectory from S1 to S0, S2, and S3. (**h**) Kaplan–Meier survival curves of integrin genes in TCGA and GEO meta-data.

**Figure 4 genes-13-02276-f004:**
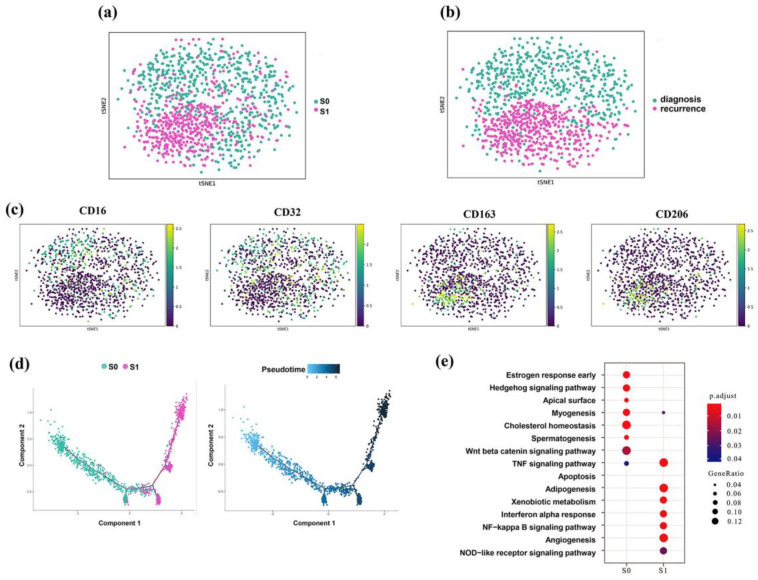
Infiltration of M2 macrophages in EOC progression. (**a**,**b**) T-SNE map of subcluster analysis for macrophages, colored by cell types (**a**) and sample groups (**b**). (**c**) Canonical markers for the M1 macrophage and M2 macrophage. (**d**) Pseudotime graph of two subgroups of macrophages inferred by Monocle 2. (**e**) Differential activated pathways among the two subgroups of macrophages.

**Figure 5 genes-13-02276-f005:**
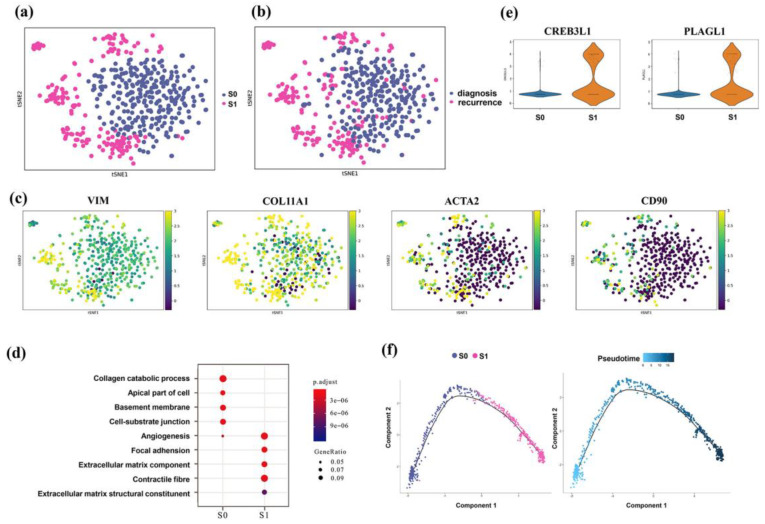
Activation of the cancer-associated fibroblast subset in EOC progression. (**a**,**b**) T-SNE view of fibroblasts, colored by cell types (**a**) and sample groups (**b**). (**c**) General stromal markers for fibroblasts and representative markers for ACTA2-positive fibroblasts. (**d**) Functional analysis of differentially expressed genes among the two subtypes of fibroblasts. (**e**) Vlnplots show the expression of genes that regulate ECM formation and components across the two fibroblast subtypes. (**f**) Pseudotime trajectory of the two subgroups of fibroblasts inferred by Monocle 2.

**Figure 6 genes-13-02276-f006:**
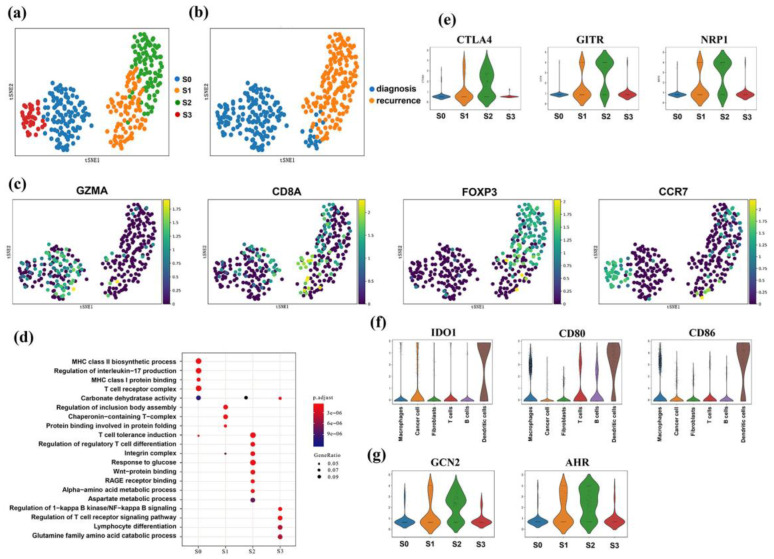
ScRNA-seq revealed heterogeneity in T immune cells. (**a,b**) T-SNE map of subcluster analysis for T immune cells, colored by cell types (**a**) and sample groups (**b**). (**c**) T-SNE maps show canonical markers for four T immune cell subtypes: GZMA for CD8^+^ T effector cells; CD8A for CD8^+^ T naïve cells; CCR7 for CD4^+^ T conventional cells; FOXP3 for CD4^+^ CD25^+^ T regulatory cells. (**d**) Functional analysis of differentially expressed genes among four subtypes of T cells. (**e**) Expression levels of other representative marker genes for CD4^+^ CD25^+^ T regulatory cells. (**f**) Dendritic cells highly expressed the IDO-related genes. (**g**) Tryptophan metabolism pathway-related genes were highly expressed in Treg cells.

**Table 1 genes-13-02276-t001:** Characteristics of patients and specimens included in this study.

Patient ID	Platform	BRCA Status	Treatment Status of Sample	Clinical Status at Time of Sampling
Patient 1	10X Genomics	WT	On-treatment	Recurrent
Patient 2	10X Genomics	WT	On-treatment	Recurrent
Patient 3	10X Genomics	N/A	Treatment-naïve	Diagnosis
Patient 5	10X Genomics	WT	Treatment-naïve	Diagnosis
Patient 6	10X Genomics	N/A	Treatment-naïve	Diagnosis

**Table 2 genes-13-02276-t002:** Cell markers for cell population identification and cell type distribution in diagnostic and recurrent samples of EOC malignant ascites.

Cell Types	Markers	Number and Ratio in All Samples	Number and Ratio in Diagnosis Samples	Number and Ratio in Recurrent Samples
Macrophages	CD68, C1QB, CD14	2104, 44.96%	1282, 48.16%	822, 40.73%
Cancer cells	PAX8, CLDN7, KLK8	806, 17.22%	278, 10.44%	528, 26.16%
Fibroblasts	COL1A2, PDPN, COL1A1,	763, 16.30%	420, 15.78%	343, 17.00%
T cells	CD2, CD7, CD3E	446, 9.53%	225, 8.45%	221, 10.95%
B cells	CD79A, CD19, CD79B	425, 9.08%	351, 13.19%	74, 3.67%
Dendritic cells	CDR7, CD83, CD1C	136, 2.91%	106, 3.98%	30, 1.49%
Total		4680, 100%	2662, 100%	2018, 100%

## Data Availability

Publicly available datasets were analyzed in this study. This data can be found here: http://www.ncbi.nlm.nih.gov/geo/, accession number: GSE 146026, GSE 14764, GSE 15622, GSE 18520, GSE 19829, GSE 23554, GSE 26193, GSE 26712, GSE 27651, GSE 30161, GSE 3149, GSE 51373, GSE 63885, GSE 65986, and GSE 9891; http://www.cancer.gov/tcga.

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
