# Peer review of "Single-Cell Sequencing of Malignant Ascites Reveals Transcriptomic Remodeling of the Tumor Microenvironment during the Progression of Epithelial Ovarian Cancer"

_genes, 2022, doi:10.3390/genes13122276_

Round 1

Reviewer 1 Report

The use of English has to be improved substantially to allow the reader to comprehend the message the authors want to convey. Also the figure quality was too poor to fully evaluated the data presented in the figures.

The study design was also not made clear. Why would you only interpret data from 5 patients if GSE14602 provides data from more patients of various different states of treatment (pre, during and recurrence)? To me this seem to be just cherry picking data in an attempt to produce new/novel data.

Author Response

Question 1: The use of English has to be improved substantially to allow the reader to comprehend the message the authors want to convey. Also the figure quality was too poor to fully evaluated the data presented in the figures.

Answer 1: Thanks for your comments. In order to improve the English language and style of the current study, we used a paid editing service to make extensive revisions in English. The changes in the revised manuscript were marked by the “Track Changes” function. I am sorry for the problem that the figures are difficult to read, we enlarged the footnotes to improve readability.

Question 2: The study design was also not made clear. Why would you only interpret data from 5 patients if GSE14602 provides data from more patients of various different states of treatment (pre, during and recurrence)? To me this seem to be just cherry picking data in an attempt to produce new/novel data.

Answer 2: As you said, GSE146026 included 22 samples from 11 patients detected by Smartseq2 and by 10x platforms, the characteristics of patients and specimens included in this study were listed in Table 1 (as follows). Among them, patients 1, 2, 3, 4, 5.1, and 6 were profiled by 10x platform, and patient 5.2, 7, 8, 9, 10, and 11 were sorted by EPCAM+CD24+ and profiled by SmartSeq2. The composition of various cell types detected on patients 1, 2, 3, 4, 5.1, and 6 were closer to the real microenvironment of ascites. In addition, patient 4 received 1 cycle of neoadjuvant were excluded from current study (Table 1). In this study, we aimed to understand the potential mechanisms of ovarian cancer from diagnosis to relapse based on scRNA-seq of acites, and we enrolled patients 1, 2, 3, 5.1, and 6 for further analysis. We added a sentence in the ‘2.1 Study subject’ subsection to explain the reason for the selection of patients at lines 78-79.

Table 1. Characteristics of patients and specimens included in this study.

Patient ID

Platform

Number of samples collected

Upfront surgery or neoadjuvant chemotherapy

Treatment status of sample

Clinical status at time of sample

Patient 1

10X

1

Upfront surgery

On-treatment

Recurrent

Patient 2

10X

2

Neoadjuvant

On-treatment

Recurrent

Patient 3

10X

1

No surgery

Treatment-naïve

Diagnosis

Patient 4

10X

1

Neoadjuvant

After 1 cycle of chemotherapy

Initial treatment

Patient 5

10X, Smartseq2

2

Neoadjuvant

Sample 1: Treatment-naïve; Sample 2: After 1 cycle of chemotherapy

Diagnosis/Initial treatment

Patient 6

10X

1

Neoadjuvant

Treatment-naïve

Diagnosis

Patient 7

SmartSeq2

5

Upfront surgery

On-treatment

Recurrent

Patient 8

SmartSeq2

3

Upfront surgery

On-treatment

Recurrent

Patient 9

SmartSeq2

3

Upfront surgery

On-treatment

Recurrent

Patient 10

SmartSeq2

1

Neoadjuvant

On-treatment

Recurrent

Patient 11

SmartSeq2

1

Upfront surgery

On-treatment

Recurrent

Reviewer 2 Report

A couple of comments to the authors.

Introduction: I would suggest the introduction to be re-written in present tense instead of past tense. It would be improved if it is more specified how many patients relapse, how many die, how the prognosis is etc. Citations are missing at certain places. As the introduction is written now, the language is seems unfinished, it is not clear to the reader how the present field is on scRNAseq on ascites in EOC, why this study is important and what was the aim of it.  

Materials and methods: Is the bulk RNAseq data separate data from the sc data? Batch effect is addressed but it is not clear in what exact way these samples actually differ (not biology but the technology and experimental run itself)

Materials and methods seem well described although beginning of Section 2.4 contains repetition and can be shortened and written more concise (rows 117-123). 

Results: Figure 1b: Show already in Figure 1 b and not just 2a what the different cell clusters are. All figures are difficult to read, the font is too small.  It is not clear why CD2 is chosen to illustrate  T-cells  in Fig 2 (instead of CD3 which is a more specific T cell marker). Some of the conclusions could be toned down a bit because the sample size is small.  

Discussion: language is non-scientific at several places. Citations are missing. The limitations with the study (limited number of samples analysed) should be discussed. 

Author Response

Question 1: Introduction: I would suggest the introduction to be re-written in present tense instead of past tense. It would be improved if it is more specified how many patients relapse, how many die, how the prognosis is etc. Citations are missing at certain places. As the introduction is written now, the language is seems unfinished, it is not clear to the reader how the present field is on scRNAseq on ascites in EOC, why this study is important and what was the aim of it.  

Answer 1: Thanks for your suggestion. We have re-written the “Introduction” section in present tense instead of past tense and added some description about “how many patients relapse, how many die, how the prognosis is etc” in the ‘Introduction’ section in the revised manuscript. We checked the whole manuscript and added references where necessary. In addition, in the “Introduction” section, we summarized the limitations of previous studies on ovarian cancer ascites and highlighted the importance of the study on the progression of ovarian cancer.

Question 2: Materials and methods: Is the bulk RNAseq data separate data from the sc data? Batch effect is addressed but it is not clear in what exact way these samples actually differ (not biology but the technology and experimental run itself)

Answer 2: The bulk RNA-seq data was separated from the scRNAseq analysis dataset. The bulk RNA-seq data and corresponding clinical information of patients with ovarian cancer were obtained from The Cancer Genome Atlas (TCGA, http://www.cancer.gov/tcga) and Gene Expression Omnibus (GEO, http://www.ncbi.nlm.nih.gov/geo/) databases. Bulk RNA-seq data retrieved from multiple databases were integrated analysis using the Combat method with the “sva” package from Python platform. The scRNAseq analysis dataset was obtained from the Gene Expression Omnibus (GSE 146026). ScRNAseq data is often compiled from multiple experiments with differences in capturing times, handling personnel, reagent lots, equipments, and even technology platforms. These differences lead to large variations or batch effects in the data, and can confound biological variations of interest during data integration. As such, effective batch-effect removal is essential [1]. In current study, the sample collection time and single-cell sequencing time were completely different among these 5 patients, and a strong batch effect might be present across different samples. To extract the biological variations of interest, we performed batch correction prior to further analysis.

References

[1] Tran H T N, Ang K S, Chevrier M, et al. A benchmark of batch-effect correction methods for single-cell RNA sequencing data[J]. Genome biology, 2020, 21(1): 1-32.

Question 3: Materials and methods seem well described although beginning of Section 2.4 contains repetition and can be shortened and written more concise (rows 117-123). 

Answer 3: Thanks for your suggestion. We have removed the repetitions to make it more concise.

Question 4: Results: Figure 1b: Show already in Figure 1 b and not just 2a what the different cell clusters are. All figures are difficult to read, the font is too small.  It is not clear why CD2 is chosen to illustrate T-cells in Fig 2 (instead of CD3 which is a more specific T cell marker). Some of the conclusions could be toned down a bit because the sample size is small.  

Answer 4: Figure 1a-b were t-SNE view of five ascites samples before DESC correction, colored by batch ID and cell clusters (unsupervised clustering results, namely, cluster 0 to 20), respectively. Figure 1c-d were t-SNE view of 5 ascites samples after DESC algorithm correction, colored by batch ID and cell clusters (unsupervised clustering results, namely, cluster 0 to 5), respectively. Figure 1d and Figure 2a looked the same, but Figure 2a were marked by specific genes of each cell type and colored by specific cell types (including Macrophages, Cancer cells, Fibroblasts, T cells, B cells, and Dendritic cells). To clarify the difference between Figure 1d and Figure 2a, we changed the “cell clusters” to “cell types” in the footnotes of Figure 2a.

I am sorry for the problem that the figures are difficult to read, we enlarged the footnotes to improve readability.We synthesized three cell markers, CD2, CD7, and CD3E displayed in Table 2 in the manuscript, to identify T cells subtypes. For the best visualization effect of T cell types, we chose a CD2-labeled T cell image in Figure 2. Due to the limitation of the sample size, we modified the description of the conclusions to make them moderate.

Question 5: Discussion: language is non-scientific at several places. Citations are missing. The limitations with the study (limited number of samples analysed) should be discussed. 

Answer 5: Thanks for your comments. We checked the “Discussion” section to make the language more scientific, and added references where appropriate. In addition, we added a review of limitations of the study.

Reviewer 3 Report

The authors report their original research regarding the remodelling of the epithelial ovarian cancer tumour microenvironment during its progression, by analysing malignant ascites via single-cell RNA-sequencing. 

The title of the paper adequately reflects its content. The abstract is well-structured, concise and highlights the most important components of the study. 

In the Introduction the authors overview the current literature regarding this specific topic. They discuss the drawbacks of bulk-cell sequencing and highlight the study by Izar et al., in which they emphasize the utility of single-cell RNA-sequencing of malignant ascites cells for the analysis of the landscape of cell composition and its interaction with the tumor microenvironment in patients with high grade serous ovarian cancer. They then clearly point out their objective to analyze the dynamic role of malignant ascites in promoting the progression of ovarian cancer.

The Materials and methods section clearly displays the study protocol. Nevertheless, I have noticed the omission of “Patient 4”, both in the text and in Table 1 - the study subjects are patients 1-3 and 5,6. In my best interest not to interfere with the author’s decision to include/omit certain patients, I believe it would be scientifically sound to explain the reason for the omission of this patient by adding a sentence addressing this matter in the ‘2.1 Study subject’ subsection.

The Results section firstly reports on the batch correction outcome, which is achieved via a deep-learning algorithm, for which a reference is provided in the Methods section. It is continued by giving an overview of the six distinct major cell clusters (macrophages, cancer cells, fibroblasts, T-cells, B-cells and dendritic cells). Each subgroup is thereafter analyzed separately, with clustered analysis of diagnostic vs. recurrent samples accompanied by appropriate figures. 

The Discussion focuses on the differences of phenotypic characteristics and biological functions in recurrent vs. diagnostic stages and use references from the literature to highlight the potential impact these differences have on cancer progression. The authors conclude by highlighting the complexity and heterogeneity of the tumor microenvironment and suggest cancer-associated fibroblasts and tumor-associated macrophages as targets for regulation of the TME - this statement lacks a more specific reasoning which would relate to the study findings. It could then be followed by a sentence commenting on more concrete future directions in this area. The Discussion section lacks a review of limitations of the study.

Author Response

Question 1: The Materials and methods section clearly displays the study protocol. Nevertheless, I have noticed the omission of “Patient 4”, both in the text and in Table 1 - the study subjects are patients 1-3 and 5,6. In my best interest not to interfere with the author’s decision to include/omit certain patients, I believe it would be scientifically sound to explain the reason for the omission of this patient by adding a sentence addressing this matter in the ‘2.1 Study subject’ subsection.

Answer 1: The present single-cell analysis dataset was obtained from the Gene Expression Omnibus (GSE 146026). GSE146026 included 22 samples from 11 patients detected by Smartseq2 and by 10x platforms, the characteristics of patients and specimens included in this study were listed in Table 1. Among them, patients 1, 2, 3, 4, 5.1, and 6 were profiled by 10x platform, and patients 5.2, 7, 8, 9, 10, and 11 were sorted by EPCAM+CD24+ and profiled by SmartSeq2. The composition of various cell types detected on patients 1, 2, 3, 4, 5.1, and 6 were closer to the real microenvironment of ascites. In addition, patient 4 received 1 cycle of neoadjuvant were excluded from current study (Table 1). In this study, we aimed to understand the potential mechanisms of ovarian cancer from diagnosis to relapse based on scRNA-seq of acites, and we enrolled patients 1, 2, 3, 5.1, and 6 for further analysis. We added a sentence in the ‘2.1 Study subject’ subsection to explain the reason for the selection of patients at lines 78-79.

Table 1. Characteristics of patients and specimens included in this study

Patient ID

Platform

Number of samples collected

Upfront surgery or neoadjuvant chemotherapy

Treatment status of sample

Clinical status at time of sample

Patient 1

10X

1

Upfront surgery

On-treatment

Recurrent

Patient 2

10X

2

Neoadjuvant

On-treatment

Recurrent

Patient 3

10X

1

No surgery

Treatment-naïve

Diagnosis

Patient 4

10X

1

Neoadjuvant

After 1 cycle of chemotherapy

Initial treatment

Patient 5

10X, Smartseq2

2

Neoadjuvant

Sample 1: Treatment-naïve; Sample 2: After 1 cycle of chemotherapy

Diagnosis/Initial treatment

Patient 6

10X

1

Neoadjuvant

Treatment-naïve

Diagnosis

Patient 7

SmartSeq2

5

Upfront surgery

On-treatment

Recurrent

Patient 8

SmartSeq2

3

Upfront surgery

On-treatment

Recurrent

Patient 9

SmartSeq2

3

Upfront surgery

On-treatment

Recurrent

Patient 10

SmartSeq2

1

Neoadjuvant

On-treatment

Recurrent

Patient 11

SmartSeq2

1

Upfront surgery

On-treatment

Recurrent

Question 2: The Discussion focuses on the differences of phenotypic characteristics and biological functions in recurrent vs. diagnostic stages and use references from the literature to highlight the potential impact these differences have on cancer progression. The authors conclude by highlighting the complexity and heterogeneity of the tumor microenvironment and suggest cancer-associated fibroblasts and tumor-associated macrophages as targets for regulation of the TME - this statement lacks a more specific reasoning which would relate to the study findings. It could then be followed by a sentence commenting on more concrete future directions in this area. The Discussion section lacks a review of limitations of the study.

Answer 2: Thanks for your suggestion. As your suggestion, we have deleted the sentence that lacked specific reasoning to make the language more scientific, and added sentences about future directions based on current explored research results. In addition, we added a review of limitation of current study.

Round 2

Reviewer 1 Report

The current version of the manuscript is significantly improved from the previous submitted version, however small changes are recommended.

Throughout the manuscript different abbreviations are used, either EOC or OC - why make this differentiation? Even within one paragraph (e.g. in discussion) both abbreviation are used for the apparently the same thing. Please choose one if same is meant or make obvious why this differentiation is made.

In the result section in multiple incidences the finding are already put into context with current literature finding, most heavily in 3.4. This type of discussing the data it intend to be done in the designated discussion section, please revise accordingly.

While the images are substantially improved the color schemes of the T-SNE figured can be altered to prove more clarity to the reader, in particular in figures 4 a and b, 5 a and b.

The explanation of how the sample inclusion criteria were set was very informative in the authors response to the reviews. However the improved manuscript version still lacks that defined clarity and can be improved upon.

Lastly, some recent studies have shown that FoxP3 can also be expressed by tumor cells. Using FoxP3 as a single selection marker for T reg cells might not be as straightforward since TC can be included in the "T reg" population (section 3.7). Can you with 100% certainty say that your population is truly T regs? Might be worthwhile viewing the data under the aspect that FoxP3 expression is reported on epithelial marker positive TC. It is a limitation of choosing single marker to differentiate particular cell populations when those marker are not only expressed by immune cells.

Author Response

Question 1: Throughout the manuscript different abbreviations are used, either EOC or OC - why make this differentiation? Even within one paragraph (e.g. in discussion) both abbreviation are used for the apparently the same thing. Please choose one if same is meant or make obvious why this differentiation is made.

Answer 1: Ovarian cancer (OC) were classified into four subtypes including epithelial ovarian cancer (EOC), germ cell tumors, sertoli cell tumor, and metastatic tumor of ovary. In current study, we explored the dynamic changes in the specific characteristics of ascites of EOC. As you suggested, we changed the “OC” to “EOC” in the place where was related to the results of present analysis, but other previous studies involving ovarian cancer that did not specify the subtype were still represented by “OC”.

Question 2: In the result section in multiple incidences the finding are already put into context with current literature finding, most heavily in 3.4. This type of discussing the data it intend to be done in the designated discussion section, please revise accordingly.

Answer 2: Thanks for your suggestion. We have moved the discussion in the “Results” section to the “Discussion” section.

Question 3: While the images are substantially improved the color schemes of the T-SNE figured can be altered to prove more clarity to the reader, in particular in figures 4 a and b, 5 a and b.

Answer 3: Thanks for your suggestion. We have changed the color schemes of the t-SNE in Figure 4a,b and Figure 5a,b to prove more clarity to the reader.

Question 4: The explanation of how the sample inclusion criteria were set was very informative in the authors response to the reviews. However the improved manuscript version still lacks that defined clarity and can be improved upon.

Answer 4: Thanks for your suggestion, we further explained how the sample inclusion in the section of “Study subject” in the revised manuscript.

Question 5: Lastly, some recent studies have shown that FoxP3 can also be expressed by tumor cells. Using FoxP3 as a single selection marker for T reg cells might not be as straightforward since TC can be included in the "T reg" population (section 3.7). Can you with 100% certainty say that your population is truly T regs? Might be worthwhile viewing the data under the aspect that FoxP3 expression is reported on epithelial marker positive TC. It is a limitation of choosing single marker to differentiate particular cell populations when those markers are not only expressed by immune cells.

Answer 5: Thanks for your comments. In current study, we explored the dynamic changes in the specific characteristics of ascites of EOC, which was an exploratory study, so we could not with 100% certainty say that our population is truly Tregs. As you said, some recent studies have shown that FoxP3 can also be expressed by tumor cells. In order to better distinguish different cell subtypes, we firstly used the cell markers in the Table 2 of the manuscript to identify different cell types (Macrophages, Cancer cells, Fibroblasts, T cells, B cells, and Dendritic cells). We then used T cells that were distinct from tumor cells to investigate the heterogeneity of T cells. We choose the FoxP3, a more specific marker in Tregs of T cells, to recognize Tregs. Based on the above principles, we identified Tregs.

Reviewer 2 Report

please see over the manuscript for types. e.g rows 136, 212, 297.  

Author Response

Question 1: please see over the manuscript for types. e.g rows 136, 212, 297.

Answer 1: Thanks for your comments. We have checked the "types" in the manuscript and modified them accordingly.